# Nanosystems as Vehicles for the Delivery of Antimicrobial Peptides (AMPs)

**DOI:** 10.3390/pharmaceutics11090448

**Published:** 2019-09-02

**Authors:** Ángela Martin-Serrano, Rafael Gómez, Paula Ortega, F. Javier de la Mata

**Affiliations:** 1Department of Organic and Inorganic Chemistry, and Research Institute in Chemistry ”Andrés M. Del Río” (IQAR), University of Alcalá, 28805 Madrid, Spain; 2Institute Ramón y Cajal for Health Research (IRYCIS), 28034 Madrid, Spain; 3Networking Research Centre on Bioengineering, Biomaterials and Nanomedicine (CIBER-BBN), 28029 Madrid, Spain

**Keywords:** AMPs, HDPs, carriers, delivery, antimicrobial agents

## Abstract

Recently, antimicrobial peptides (AMPs), also called host defence peptides (HDPs), are attracting great interest, as they are a highly viable alternative in the search of new approaches to the resistance presented by bacteria against antibiotics in infectious diseases. However, due to their nature, they present a series of disadvantages such as low bioavailability, easy degradability by proteases, or low solubility, among others, which limits their use as antimicrobial agents. For all these reasons, the use of vehicles for the delivery of AMPs, such as polymers, nanoparticles, micelles, carbon nanotubes, dendrimers, and other types of systems, allows the use of AMPs as a real alternative to treatment with antibiotics.

## 1. Introduction.

Infectious diseases were considered, at the beginning of the twenty-first century, as one of the most important causes of death in humanity, even though their relative percentage has been decreasing since the nineteenth century [1]. During the 1940s, the introduction of antibiotics into clinical practice was one of the most important advances for their treatment, and increased the life expectancy for several years [2]. Antibiotics have saved millions of lives, and in addition, they have revolutionized medicine [3]. They have also played a highly significant role in progress in a number of fields, such as solid organ transplants and hematopoietic progenitors, survival of premature and immunocompromised patients (natural or by pharmacological therapies), and surgery of prosthetic material and vascular catheters, where infections are especially prevalent and potentially important [4,5]. However, for several years now, a growing threat has deteriorated the efficacy of these drugs: Bacterial resistance to antibiotics that causes prolonged hospital stays, incurring high medical costs and causing more mortality, making the global health issue one of the greatest threats. For that, it is crucial to develop new drugs to overcome the resistance acquired by microorganisms against conventional antibiotics [6,7].

On the lookout for new approaches to antibiotic resistance, antimicrobial peptides (AMPs) are currently presented as a promising therapeutic solution, since they have a wide spectrum of activity against several pathogenic microorganisms and can be considered as natural antibiotics [8,9]. They are key effector molecules in the innate immunity of organisms, isolated from mammals, birds, amphibians, crustaceans, fish, insects, plants, and microorganisms. AMPs are predominantly small cationic peptides with hydrophobic regions containing between 10–100 amino acids, especially arginine residues, which allows them to interact with negatively-charged membranes, causing the direct destabilization of the surface of membranes with pore formation and subsequent cell lysis [10,11]. Also, they have been described as chemotactic agents [12], modulating the immune system, and therefore constituting a bridge between innate immunity and adaptive immunity [13] (Figure 1).

However, despite the multiple beneficial properties of AMPs, they present some disadvantages such as: (i) Degradation by proteases, both in the bloodstream and in the gastrointestinal system; (ii) their union with others proteins, which leads to their inactivation; (iii) low metabolic stability and oral absorption; (iv) rapid excretion through kidneys and liver; (v) high toxicity and immunogenicity; and (vi) high production costs. For these reasons, their use for in vivo applications have not been fully satisfactory and only a few of them were explored in clinical trials [14].

Recent studies have focused on nanotechnology in order to minimize drawbacks of natural and synthetic AMPs. Nanotechnology is the term given to those areas of science and engineering where phenomena taking place at dimensions in the nanometric scale are utilized in the design, characterization, production, and application of materials, structures, devices, and systems [15,16]. Currently, it has attracted the attention of a large number of research groups due to the enormous advantages that its use present in different fields, and more specifically in biomedicine [17,18]. One of the main goals of nanotechnology is the design of nanocarriers, promising biomaterials that could increase therapy efficacy, minimize side-effects, and offer a controlled pharmacokinetic profile, as well as direct administration towards the target organ, protect the encapsulated peptide from degradation, and reduce toxicity. Diverse types of nanomaterials, including polymers, liposomes, hydrogels, self-assembly systems formed by surfactants, (block co)polymers, and polar lipids polymer (micro)gels, as well as wide range of inorganic nanoparticles/nanomaterials, each offering system-specific opportunities, have been explored as delivery systems not only in the transport of peptides [19], but also for gene therapy [20], cancer treatment [21], and drug delivery [22].

The present review is focused on different delivery systems for AMPs (see Table 1 and Figure 2), describing several systems reported in bibliography during the last five years. The classification of these nanosystems as AMP delivery vehicles has been made based on the nature of their skeleton: Organic or inorganic. Furthermore, we reviewed AMPs as vehicles themselves. 

## 2. Inorganic Nanosystems

### 2.1. Metal Nanoparticles 

Nanoparticles (NPs) are particles in the size range of 0.1 to 100 nm and exhibit completely innovative physicochemical properties in comparison with their bulk counterparts [32]. Depending on the shape, dimensions, and properties, it is possible to classify them in different manners [33]. One of the most commonly used systems for AMP transport is that of gold nanoparticles (AuNPs). Galdiero et al. [34] (2018) described the synthesis of AuNPs funtionalizated with indolicidin AMP, employing thiol chemistry to covalently attach the peptide to AuNPs. The peptide/AuNP conjugate presented a higher efficacy in preventing cell adhesion and destroying biofilms formed by *Candida albicans* in in vitro experiment compared to AuNPs and indolicidin alone, probably due to the fact that the peptide is protected from degradation by the proteases. Mangoni et al. [35] (2017) reported the first example of a covalent bond between the AMP esculentin-1a, with high activity against *Pseudomonas aeruginosa* bacteria, and soluble AuNPs through polyethylene glycol (PEG) linker. The conjugate AuNPs@Esc(1-21) presented close to 15-fold the antipseudomonal activity of the Esc(1-21) alone and did not present toxicity in human cells. In addition, this conjugate demonstrated to be stable after conjugation with the AuNP, keeping their activity over months. The mechanism of action of these type of AuNP-AMP conjugates has been discussed by various authors [36,37]. The most commonly accepted mechanism is bacterial cell membrane rupture caused by the interaction of the nanoparticle conjugate with the negatively-charged components of the membrane, without intracellular entry of the conjugate. 

The use of silver nanoparticles (AgNPs) enables to combine the widely known antibacterial effect of silver with the effects of AMPs. Some studies have revealed that peptides modified with cysteine moieties increase conjugate stability [38,39]. One of the latest works published in bibliography by Bhunia et al. (2019) [40] studied the functionalization of AgNPs with potent Andersonin-Y1 peptide against the multidrug resistant strains *Klebsiella pneumoniae*, *Pseudomonas aeruginosa,* and *Enterobacter* species (experimentally, it was estimated that approximately ~200 peptides coated the surface of the nanoparticle). Again, the antimicrobial effect of the AY-AgNP conjugate was more than the sum of the activities of the peptide and the nanoparticle taken separately. In order to elucidate the mechanism of action, several NMR studies (in real time), together with molecular dynamics studies, fluorescence-based dye-leakage and hemolytic assay, were carried out. All these studies showed that interaction with hydrophobic tails of the bacteria membrane causes pores, favoring that AgNPs cross the barrier and cause cell death by attaching to its DNA.

### 2.2. Carbon Nanotubes 

Carbon nanotubes (CNTs) can be divided in two classes: The so-called single-walled (SWCNT) or the multi-walled (MWCNT), both of them formed by rolled-up tubular shells of graphene, and presenting physical properties that offer great value for the development of advanced biomaterials [41]. Aich et al. (2015) conjugated indolicidin AMP to CNTs and AuNPs funcionalizated with carboxylic acid on the surface, using EDC-NHS conjugation protocol [42]. In this paper, they characterized both conjugates by different techniques and compared their properties, finding that both induced complementary innate immune gene activation. CNT-indolicidin might also protect host cells towards bacterial infection at 1000-fold less concentration than free indolicidin. Furthermore, the gene expression profile of indolicidin was different depending on the carrier, the use of CNT presented or activated more pro-inflammatory genes, while the AuNP conjugate activated Il-10, a gene with anti-inflammatory function. Most recently, Chaudhari et al. (2019) have analyzed the toxicity and antimicrobial activity of different AMPs (TP359, TP226, and TP557) supported in silver-coated CNTs against *Staphylococcus aureus* infection using a full thickness human 3D skin model [43].

On the other side, Koksharova et al. (2018) described the use of CNTs to remove arenicin-1 and tachyplesin-1 AMPs from aqueous solutions [44]. The nanotubes used to carry out this work were Taunit-Mb [45]**,** functionalized with –COOH moieties in their structure, that efficiently adsorbed peptides containing free –NH_2_. The comparison of results obtained with a conventional absorbent, showed that the amount of peptide found in CNTs was three times more than in conventional material, probably by formation of ionic bonds between AMP-CNTs. Table 2 summarize the most relevant information about Inorganic Nanosystems described in this section.

## 3. Organic Nanosystems

### 3.1. Polymer Systems 

#### 3.1.1. Polymers

Polymers are macromolecules containing several repeating units of a smaller molecule (monomer). There are many natural polymers such as DNA, cellulose, or chitosan, and many others, such as poly(lactide-*co*-glycolide) (PLGA) or PEG, are synthetic. The use of biodegradable/biocompatible polymers in biomedicine and the food industry has increased in the last decades. During 2017–2018, some review articles, including state-of-the-art polymer-based strategies to improve in vivo biocompatibility and delivery systems for AMPs, were published [10,46,47]. Furthermore, in the following paragraphs, we gather an update on the most recently published works reporting the use of polymer nanoparticles [48,49,50], nanofibers [51,52], multilayers [53,54], polymer-coated surfaces [55,56], and polymer conjugates [57,58,59,60,61,62,63] for AMP delivery. 

First, we present three very recent examples of polymer nanoparticles use for AMP delivery. Casciaro et al. (2019) developed PLGA NPs for the delivery of esculentin-1a-derived AMPs in cystic fibrosis patients presenting *Pseudomonas aeruginosa* infection, and observed an improved vehiculization and efficiency inhibiting bacterial growth [48]. In the same year, Vijayan et al. (2019) investigated the wound-healing potential of PLGA NPs carrying two growth factors entrapped in their interior, and the K4 AMP covalently conjugated, which showed a sustained release and an improved cell proliferation, as well as broad-spectrum antimicrobial activity [49]. Previously, Almaaytah et al. (2017) encapsulated the RBRBR ultrashort AMP into chitosan NPs. This system reduced toxicity compared to the free AMP and increased antibacterial activity compared with the unloaded chitosan NPs [50].

Nanofibers are another kind of polymeric material recently used as nanocarriers for AMPs. Soto et al. (2019) prepared amaranth protein isolate: Pullulan (API-PUL) nanofibers loaded with nisin AMP for food safety applications. The release behavior of nisin from API-PUL nanofibers resulted to be progressive and show bactericidal activity against *Salmonella typhimurium*, *Listeria monocytogenes,* and *Leuconostoc mesenteroides*, which evidenced the protection of nisin antimicrobial activity while in contact with food [51]. The previous year, Amariei et al. (2018) incorporated the AMP ε-poly(l-lysine) (ε-PL) to poly(acrylic acid) (PAA)/poly(vinyl alcohol) (PVA) electrospun nanofibers for potential use in biomedicine, and observed a considerable decrease in bacterial growth, compared to non-AMP-loaded PAA-PVA, against *Staphylococcus epidermidis*, *Staphylococcus aureus*, *Escherichia coli* [52].

The design of multilayer-based materials has also been explored. He et al. (2018) prepared a layer-by-layer microsphere-loaded nanofiber membrane with antibacterial activity for bond regeneration. The methodology consisted of first electrospinning a gelatin (Gln) and chitosan (CS) composite containing hydroxyapatite nanoparticles (nHAp), and then electrospraying short tryptophan-rich AMP Pac-525 PLGA microspheres (AMP@PLGA-MS). The membrane showed to be biocompatible and be able to promote osteoblasts differentiation, as well as one-week bactericidal activity and up to one-month antibacterial activity against *Staphylococcus aureus* and *Escherichia coli* [53]. Very recently, Rodríguez López et al. (2019) coated titanium surfaces with chitosan/hyaluronic acid polymer multilayers for local delivery of AMP β-peptide mimetics to prevent bacterial biofilm generation in orthopedic implants, and observed not only a controlled release of the antimicrobial that prevented biofilm formation for 24 days and five bacteria, but also no toxicity toward preosteoblasts [54].

Regarding the use of surfaces, Xiao et al. (2018) compared the stability over time of two polymer surfaces chemically decorated with cecropin–melittin hybrid AMP, and observed that the stability of the surface prepared by chemical vapor deposition polymerization was higher than that of the assembled monolayer, the former being more suitable for applications in antimicrobial coating [55]. Muszanska et al. (2014) prepared an antiadhesive polymer brush coating for biomedical devices by the conjugation of both AMPs and arginine–glycine–aspartate (RGD) peptides to the PEG chains of the triblock copolymer Pluronic F-127. As a result, they obtained a multifunctional surface with not only antiadhesive and bactericidal properties, but also with cell growth capacity [56]. 

AMP conjugation to polymers not only preserves their antimicrobial activity, but also reduces their toxicity and provides new functionalities [57]. Sun et al. (2018) last year reviewed the synthetic methodologies that have been used to date to create AMP–polymer conjugates and biomedical applications of these systems [57] and, in the following lines, we present some examples of the research in AMP conjugates not included in Sun review [58,59,61]. Gong et al. (2017) reported that arginine selective PEGylation of arginine rich AMPs reduces inherent AMP toxicity, confers protection against serum proteases, and allows the steady release of the AMP bioactive form [58]. Kelly et al. (2016) showed that bioreversible PEGylation of AMPs could increase the potential of these antimicrobials in cancer therapy [59]. Although PEGylation has been widely used for peptide drug conjugation, e.g., magainin 2, tachyplesin 1, and nisin, among previously PEGylated AMPs [62], the immunogenicity observed in some cases—together with the low number of groups that allows conjugation—has led to the use of alternative polymers such as biocompatible hyperbranched polyglycerol (HPG). HPG is easy to synthesize, presents long blood circulation, and has several hydroxyl groups that can be functionalized [61,62]. Kumar et al. (2015) first reported HPG conjugation to an AMP [62] and more recently studied AMP-HPG conjugates generated with different numbers of aurein 2.2-derived peptides to increase their antimicrobial activity and decrease their toxicity [63]. Abbina et al. [61] recently reviewed advances in synthesis, biocompatibility, and biomedical applications of HPGs. An overview about the most relevant information about the polymers described is presented in Table 3.

#### 3.1.2. Hydrogels

Hydrogels can be defined as water-swollen networks of polymers, formed by cross-linking of hydrophilic polymer chains within an aqueous microenvironment. They present unique properties and can be loaded with different types of molecules for a variety of applications [64,65].

Liskamp et al. (2014) describe one of the first examples with hydrogel networks based on cross-linked PEG diacrylate-based (PEGDA) for AMP vehiculization using thiol−ene photoclick chemistry in a single-step procedure [66]. AMP-hydrogels exhibited great antimicrobial activity against Gram-positive *Staphylococcus aureus* and *Staphylococcus epidermidis* and Gram-negative *Escherichia coli* in vitro. Since then, there are many papers showing several kinds of hydrogels as AMP carriers. Mello et al. (2016) [67] immobilized the Cecropin A over PEGDA hydrogel cores using diverse molecular linkers, containing thiol moieties, between peptide and hydrogel. The –SH functions present in the linker reacted with maleimide groups located in the modified peptide structure. Their results in *Escherichia coli* bacteria showed that the antibacterial activity of hydrogel-AMP conjugate was dependent on the linker size and the amount of peptide loaded in the hydrogel. Malkoch et al. (2018) [68] investigated how the charge density in Poly(ethyl acrylate-*co*-methacrylic acid) or poly(ethyl acrylate (EA)/methacrylic acid (MAA)/1,4-butandiol diacrylate (BDDA)) microgels (MAA26.5 and MAA60 microgels) affects the capacity to release the peptides AP114, DPK-060, and LL-37 effective against *Pseudomonas aeruginosa* and *Escherichia coli* bacteria, and studied hemolysis, proteolytic stability, and interaction of loaded hydrogel with membranes [68]. Table 3 shows summarize some of the most important properties described for hydrogels as delivery systems for AMPs.

### 3.2. Lipid-Based Systems

#### 3.2.1. Liposomes 

Liposomes are nanometric hollow spherical artificial vesicles consisting of one or several concentric lipid bilayers and are formed by cholesterol and phospholipids surrounding an aqueous cavity [69,70,71,72]. The amphiphilic character of phospholipids allow their self-assembly with the polar head groups oriented to the inner and outer aqueous phases, and the fatty acid hydrophobic tails oriented to the bilayer interior [73]. 

Bangham et al. first described liposomes in 1965, and Gregoriadis et al. reported their first use as nanovehicles in 1971 [69]. Since then, many cases of liposomes as potential candidates for drug delivery [72,74,75], diagnostics [76,77], and theragnostics [65,72,78,79,80] have been reported for numerous fields of applications such as antibiotics, anticancer, and gene therapy [71]. Furthermore, some drugs based on liposome formulation have been commercialized or are in clinical trials [72,81,82,83], among which, some examples are Exparel^®^ (anesthetic), DepoCyt^®^, DaunoXome^®^, Myocet^®^, Caelyx^®^/Doxyl^®^, Mepact^®^ and Marqibo^®^ (anti-cancer), DepoDur^®^ (pain relief), and Visudyne^®^ (macular degeneration, myopia, degenerative) [82].

The broad range of liposome applications can be attributed to their biocompatibility, along with their ability to encapsulate both hydrophobic and hydrophilic molecules inside the lipid bilayer or in the aqueous cavity, respectively [71,81,84]. Furthermore, this encapsulation prevents in vivo decomposition of cargo molecules, either via enzymatic or chemical degradation, or via immunological neutralization [81,84], as well as reduces unspecific delivery [71]. Composition and size of liposomes can be tailored to vehiculize a specific molecule, decrease their rate of degradation, control the release, or even increase their affinity for a specific target [69,71]. For all that, encapsulation of AMPs into liposomes is a desirable strategy to prevent the drawbacks associated with the direct application of these AMPs, since cytotoxicity could be decreased and stability and bioactivity enhanced [85]. Thus, the number of publications researching the antimicrobial activity of AMPs encapsulated in liposomes, in fields such as food technology [86,87,88] and biomedicine [47,85,89], has risen during the last few years (Table 4).

As a recent example, Cantor et al. (2019) [86] carried out structural modification of the AMP Alyteserin-1c and coated liposomes with Eudragit E-100^®^, a nontoxic cationic polymer approved by the FDA Inactive Ingredients Guide, as a nanovehiculization system for effective alternative antimicrobials applications in food safety. As a result, they observed an increase in antibacterial activity of approximately 2000 times against *Listeria monocytogenes* and 12.5 times against *Escherichia coli*, in comparison with the unencapsulated peptide.

In previously reported works, Lopes et al. (2017) [87] and da Silva et al. (2014) [89] encapsulated nisin in phosphatidylcholine liposomes coated with biocompatible polysaccharides such as pectin [87], polygalacturonic acid [87], chitosan [89], or chondroitin sulphate [89] and evaluated their efficiency in inhibiting *Listeria innocua 6a, Listeria monocytogenes ATCC 7644, Listeria monocytogenes 4b, Listeria sp. Str1,* and *Listeria sp. Str2* for food safety applications. In the former study, in vitro release studies for polysaccharide-coated liposomes resulted to be lower when compared with uncoated liposomes [87], evidencing a potential use for food application. In the latter, the formulation containing chitosan was more stable and more efficient for inhibiting *Listeria monocytogenes* when compared with the chondroitin sulphate-coated liposomes [89]. Also, Pu et al. (2016) [88] reported the antibacterial and anti-biofilm properties of chitosancoated liposomes encapsulating Apep10. These liposomes presented increased stability compared with the uncoated analogues, which is in concordance with da Silva findings [89], and indicates that the surface coating inhibited undesirable aggregation and peptide release during storage. Another example is the work of Ron-Doitch et al. (2016) [85], who prepared LL-37 and indolicidin liposomes coated with PEG and evaluated their activity against herpes simplex virus 1 (HSV-1). They found lower toxicity and enhanced antiviral activity for LL-37 liposomes compared with both the free AMP and indolicidin liposomes. Gomaa et al. (2017) demonstrated that liposome-encapsulated AMP microcin J25 presented an effective protection against degradation during gastrointestinal digestion when dual coated with the biopolymer pectin and whey proteins [90]. 

#### 3.2.2. Liquid Crystalline Particles 

Liquid crystalline nanoparticles (LCNPs), such as cubosomes and hexosomes, are nanostructured liquid crystalline particles that consist of lipid bilayers that fold to acquire two- and three-dimensional structures with interwoven water channels [91]. They are more thermally stable than other carrier systems used for drug delivery such as liposomes, niosomes, or micro sponges [92]. One of the most commonly used lipids for LCNP manufacture is the biodegradable lipid glyceryl monooleate (GMO), and diverse AMPs have been successfully loaded in cubosome or hexosome form. One of the first examples of cubosome use as an AMP carrier was described by Anderson et al. (2016) [93,94]. In this work, they studied cubic LCNP post-load with AP114, DPK-060, and LL-37 AMPs using different conventional formation routes, and studied the preservation or improvement of the peptides antibacterial effect. Moreover, Boge et al. (2019) explored the cargo of LL-37 into GMO-based cubosomes using three different manners: Pre-loading, post-loading, and hydrotrope-loading (incorporation during cubosome generation) [95]. The results show that the strategy employed to incorporate the AMP determines their structure and leads to variations of peptide concentration into the cubosome. Chorilli et al. (2016) used liquid crystalline systems formed by tea tree oil, polyoxypropylene-(5)-polyoxyethylene-(20)-cetyl alcohol, and polycarbophil polymer dispersions as the aqueous phase to load the AMP p1025. This formulation was proven to be effective in the treatment of dental caries, showing that polymer dispersions favor adhesion onto the teeth [96]. The Table 4 includes the most significant results based on the use a liquid crystalline particles described as carriers of AMP.

### 3.3. Dendritic Systems

Dendrimers are highly-branched, star-shaped macromolecules with nanometer-scale dimensions. They are obtained from controlled synthetic routes that lead to the formation of monodisperse systems characterized by high structural precision with multiples end-groups, which can be modified to modulate their physicochemical or biological properties. Depending on the topology of the system, could find spherical systems, dendritic wedges, janus, or bow-tie dendrimers and attending to the chemical structure of the skeleton, the most common are polyamidoamine (PAMAM) [97], polypropyleneimine (PPI) [98], poly-l-lysine (PLL) [99], polyglycerol (PG) [100], poly(bencyl ether) [101], carbosilane [102], or phosphorous dendrimers [103]. In spite of the fact that dendrimers have been employed as suitable delivery systems for different kinds of molecules [27,104], only a few examples can be found in bibliography describing them as carriers for AMPs. 

Our research group (2019) has recently described different generations of cationic carbosilane wedges functionalized with a maleimide group in the focal point that could attach different AMPs containing a cysteine residue in their structure, and studied covalent and no covalent dendron-AMP systems [105]. In this work, it was shown that there exists a synergy effect in antibacterial activity against *Staphylococcus aureus* and *Escherichia coli* when the dendron of second generation is conjugated with an AMP. In addition, the studies carried out demonstrate that dendrons, AMPs, and their covalent conjugates have the capacity to pass through the membrane, causing morphological damage as well as deteriorating the cellular integrity of the membrane, and the mechanism of action probably seems different for the AMP or dendrimer alone and conjugate (Table 5).

### 3.4. Cyclodextrins 

Cyclodextrins are cyclic oligosaccharides composed by several dextrose units attached by means of α-1,4-glucosidic bonds [106]. They have a hollow structure, presenting a hydrophobic inner cavity and a hydrophilic surface [107], which confers them the ability to form supramolecular complexes with a variety of molecules, increasing their stability, biocompatibility, and solubility [106,107,108]. For that reason, cyclodextrins have been used in formulations for the pharmaceutical, cosmetic, and food industries [106,107]. 

In recent years, the use of cyclodextrin for AMP delivery has been reported by several authors, since hydrophobic residues of AMPs can be inserted in the hydrophobic cavity of cyclodextrins. Teixeira et al. (2016) complexed KR12 AMP with 2-hydroxypropyl-β-cyclodextrin and evaluated its antimicrobial and antiproliferative properties for cancer therapy improvement. They observed that the complex inhibited both bacterial growth and fibroblast proliferation, as well as reduced hemolysis [108]. Li et al. (2017) demonstrated that the formation of an inclusion complex between CM4 AMP and β-cyclodextrin not only increased storage stability and protection against proteinases, but also the antibacterial activity remained the same, having the system a great potential in food industry [106]. Zhang et al. (2018) encapsulated alamethicin into γ-cyclodextrin and observed higher solubility and temperature and pH stability compared to the AMP alone, as well as efficient antimicrobial activity against *L. monocytogenes*, being the minimum inhibitory concentration (MIC) of γ-cyclodextrin/alamethicin molar ratio dependent [107] (Table 5).

### 3.5. Aptamer Conjugates

Nucleic acid aptamers, also known as chemical antibodies, are RNA or single stranded DNA molecules, composed by 20–80 nucleotides, able to bind specifically and with high affinity to pre-selected target molecules thanks to their unique three-dimensional structure. They are obtained using the Systematic Evolution of Ligands by Exponential Enrichment (SELEX) system. The fact that aptamers are produced without need for animal experimentation, their smaller size compared with antibodies, and their easier modification, are some of the advantages that have made them a promising tool in the biomedical field, mainly for biosensing or for targeting molecules for therapy, imaging, and drug delivery [109,110].

In recent years, it has been reported the use of AuNPs conjugated with DNA aptamers for intracellular delivery of AMPs into mammalian cells and the evaluation of their suitability as potential treatment of intracellular bacterial infections in mammals [111,112]. Yeom et al. (2016) conjugated AuNPs with histidine-tagged DNA aptamer and further decorated them with C-terminally hexahistidine-tagged A3-APO. Their application to HeLa cells infected with *Salmonella Typhimurium* eliminated intracellular bacteria, increasing cell viability. Moreover, in vivo experiments performed in mice resulted in the inhibition of bacterial colonization of mice organs and 100% animal survival [111]. Lee et al. (2017) obtained similar successful results against *Vibrio vulnificus* infection with HPA3P AMP loaded on AuNPs conjugated to a DNA aptamer [112]. All these results are collected in Table 5.

## 4. AMPs as Vehicles

Although we have focused so far in the delivery systems for AMPs, it has been reported that AMPs can also act as delivery vehicles [113,114] for bioactive compounds [115] or liposomes [116,117]. 

Hu et al. (2019) studied the use of indolicidin as a vehiculization system for oligodeoxynucleotides (ODNs). They created indolicidin dimers (LIC and CIL) in order to increase AMP charge density, which were evaluated as vehicles of ODNs against tumor necrosis factor α (TNF-α). CIL showed good vehiculization capacity, silencing TNF-α expression for more than 14 h, a result that makes CIL complexes a promising tool for oligonucleotides delivery with application in gene silencing [115]. In addition, in previous years, a couple of works using AMPs as vehicles for targeted drug delivery systems were published [116,117]. Mizukami et al. (2017) developed versatile stimuli responsive controlled release systems based on the combination of modified temporin L (TL) with surface-anionic liposomes. TL modifications consisted of: (i) The substitution of a Lys residue in TL for the protease-triggered system, or (ii) the substitution of a neutral amino acid for an anionic phosphorylated amino acid in the TL lipophilic region for the phosphatase-triggered system [116]. Zhang et al. (2016) reported a vehiculization system for tumor delivery based on liposomes surface functionalization with the pH responsive AMP [D]-H6L9 [117].

The design of hybrid peptides in which an AMP able to permeabilize membranes is combined with another AMP able to act intracellularly after translocating across the membrane, can be a breakthrough in the improvement of antibacterial activity of current AMPs. The study of the order or the influence of linkers between the two peptides are key factors to be investigated during the development of hybrid AMPs [118]. Wade et al. (2019) recently developed and evaluated a library of sixteen hybrid peptides, giving rise to interesting and promising results about the dominating involved mechanism and the order to reach an improved antibacterial activity of hybrid peptides, as well as highlighting the fact that the use of AMP cocktails could also have great potential for improving AMP efficacy [118].

More complex peptidic systems are the so-called antimicrobial peptide dendrimers (AMPDs), based on branched polymers bearing several peptides covalently attached to a core [119,120]. These multivalent peptide mimetics have gained attention over the last few years due to their increased antibacterial activity, high resistance to degradation, probably due to a major steric hindrance, and the higher concentration of bioactive units per dendrimer molecule [119,120,121]. Scorcipiano et al. published a nice complete review in which several examples of AMPDs reported until 2017 are gathered [122]. More recent examples of the effective use of AMPDs as antibacterial agents are presented in the following lines. Siriwardena et al. (2019) developed dendrimeric peptides by the combination of elements from different AMPDs. The combination of G2 dendrimer TNS18 peripheral branches with the inner branches and core of G3 dendrimer T7 gave the novel AMPD DC5, whose activity is the combination of those of the parent AMPDs. Thus, this strategy was demonstrated to have great potential in the development of improved AMPDs for fighting against multidrug resistance [123]. Grassi et al. (2019) evaluated in vitro the antibiofilm efficacy of the semisynthetic peptide lin-SB056-1 in comparison with its dendrimeric derivative (lin-SB056-1)2-K, demonstrating for the latter increased biofilm inhibition and lower cytotoxicity [124]. In a near future, the combination of AMPDs with any of the nanovehicles previously reported in this review could make even more effective and promising the use of these multimeric structures for beating multidrug resistant infections. 

## 5. Conclusions

AMPs, also known as HDPs, have high potential as new therapeutic agents whose ability to kill bacteria depends on how they interact with bacterial membranes or cell walls. Different nanosystems, among which inorganic nanoparticles, synthetic polymers systems, liposomes, dendrimers, cyclodextrins, and aptamers are included, have been employed as carriers for AMPs with the aim to make feasible the use of these new antimicrobial agents for food safety or biomedicine. The way nanovehicles are coupled to the AMP varies from covalent attachment in the case of nanoparticles, polymers, or surfaces, to supramolecular complexation in the case of cyclodextrins—or even encapsulation, as it happens in many liposome formulations. It is important to highlight that the combination of AMPs with these nanovehicles allows not only to protect AMPs from degradation, improve their solubility, decrease their cytotoxicity, and even broaden their antibacterial activity compared with the AMP alone, but may also allow their delivery to a desired target. 

In the near future, the trend in the design of more effective AMPs seems to be directed to the development of hybrid AMPs in which a membrane permeabilizing peptide is combined with another peptide able to act intracellularly. Also, multivalent AMPs or AMPDs have been demonstrated to possess improved properties compared with their monovalent counterparts, so their further vehiculization using any of the nanocarriers currently available could better their stability, biocompatibility, and efficacy as antimicrobials.

In conclusion, AMP vehiculization seems to be on the way to becoming the strategy to be used for making transferable to clinical practice the administration of these new antibacterial agents, which would be an alternative to the prescription of conventional antibiotics and could avoid the problems related with antibiotic resistance, either for healthcare costs or for patient quality of life. 

Thus, nanotechnology may be a very effective tool in the near future to move AMPs from being a promising alternative in the laboratory to a reality in clinical practice, where the need to find new compounds with antibiotic capacity is already a health problem of concern in most developed countries, where processes of resistance to traditional antibiotics are becoming more frequent. To this end, it will be necessary to design new nanosystems or improve the existing ones in order to convey—in a precise and targeted way—the AMPs to their site of action, at the same time as deepening the study of the mechanism of these nanoconjugates in their antibacterial activity.

## Figures and Tables

**Figure 1 pharmaceutics-11-00448-f001:**
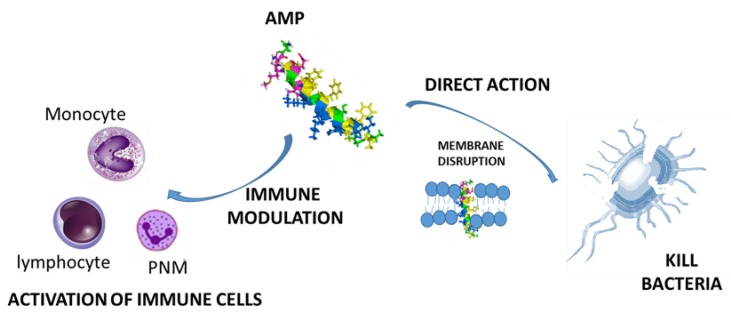
Antimicrobial peptides’ (AMPs) mechanism of action.

**Figure 2 pharmaceutics-11-00448-f002:**
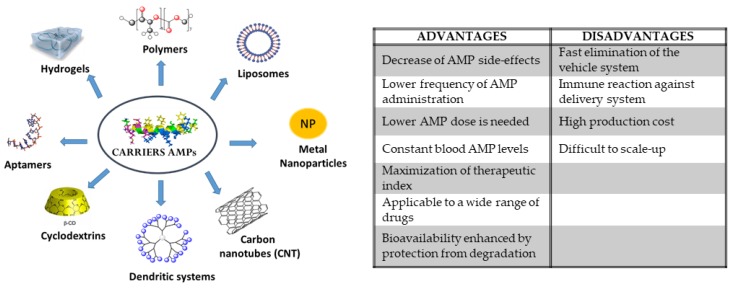
Some examples of AMP carriers and some advantages and disadvantages of their use.

**Table 1 pharmaceutics-11-00448-t001:** Advantages and disadvantages of different nanostructures as carriers.

Drug Carrier	Advantages	Disadvantages
Metal Nanoparticles [23]	- Multimodal applications - High surface area	- Metal Toxicity - Stability - Storage
Carbon nanotubes [24]	- Water soluble - Multifunctional	- Expensive to produce - Low degradability
Liposomes [25]	- Biodegradable - Load hydrophobic and hydrophilic drugs	- Poor drug loading - Immunogenicity
Liquid crystalline particles [26]	- Thermally stable - Biodegradable - Bioadhesion	- Complex and difficult to prepare - Low encapsulation rate
Dendritic systems [27]	- Monodisperse molecules with a high control over the critical molecular design parameter	- Highly expensive synthesis processes - Non-specific toxicity
Polymers [28]	- Biocompatible - Biodegradable (depending on polymer nature) - Easy to modify - Controlled drug release	- Low cell affinity - Toxicity of degradation products
Hydrogels [29]	- High water content and biocompatibility - Could be gelling at body temperature	- Trend to be fragile - Expensive, especially the smart hydrogels - Behavior difficult to predict
Cyclodextrins [30]	- High aqueous solubility - Chemical stability	- Could be irritant
Aptamers [31]	- Good chemical stability - Isotropic properties - Unlimited shelf life - Highest specificity and affinity to the target	- Unpredictable risk - Complex and costly procedures

**Table 2 pharmaceutics-11-00448-t002:** Overview of the different inorganic nanosystems as delivery systems for AMPs.

Peptide/ Antibacterial Activity	Sequence	Delivery System	Bacterial Strain	Findings
**Indolicidin** **Broad spectrum**	CILPWKWPWWPWRR	AuNPs [34]	*C. albicans*	Biofilm formation inhibition at 24 h was 40% for indolicidin alonem and more than 50% for AuNPs–indolicidin. Eradication of 48 h biofilms with AuNPs–indolicidin was 55–65%.
**Esculentin-1a**	GIFSKLAGKKIKNLLISGLKG	AuNPs [35]	*P. aeruginosa*	AuNPs-Esc(1-21) preserved a concentration-dependent microbicidal effect, and killing activity was ∼12-fold increased since MBC_50_ was reduced from 1 for Esc(1-21) alone to 0.08 μM. Also, AuNPs-Esc(1-21) kept their antibacterial activity in the presence of trypsin. Unlike for the peptide alone, AuNPs-Esc(1-21) produced bacterial death by a membrane perturbation mechanism.
**Odorranain-A-OA1**	VVKCSYRLGSPDSQCN	AgNPs [38]	*E. coli*	Bacterial death increased to 60% for the AgNP-OA1 conjugate, while it was 31% and 33% when only AgNP and peptide were added, respectively, and ∼30% when AgNP and peptide were added together but not conjugated. Conjugate cytotoxicity was evaluated in HaCaT cell line, and a higher biocompatibility (no significant IC_50_ values) was found compared with AgNP alone, with IC_50_ value of ∼96 μg/mL.
**Andersonin-Y1 (AY1)** **CAY1** **AY1C**	FLPKLFAKITKKNMAHIR CFLPKLFAKITKKNMAHIR FLPKLFAKITKKNMAHIRC	AgNPs [40]	*K. pneumoniae,* *P. aeruginosa,* *S. typhimurium*	The activity of the AgNP peptide was more than the sum of the activities of the peptide and the nanoparticle taken separately. The mechanism of action was alteration of bacterial cell surface morphology followed by membrane rupture.
**Indolicidin** **Broad spectrum**	CILPWKWPWWPWRR	CNTs [42]	*S. typhimurium*	CNT conjugated indolicidin at 0.02 μg/mL protected the cell from challenge of the bacteria significantly better than free indolicidin at 20 μg/mL.
**TP359** **TP226** **TP55**	MYRKKALKKD	SWCNTs-Ag [43]	*S. aureus*	In all cases, the conjugates presented a slight improvement of MIC where the nanotube was cargo whit 5 mg/mL of AMPs over human skin model.

MIC: Minimum inhibitory concentration; IC_50_: Half maximal inhibitory concentration; EC_50_: Half maximal effective concentration; EC_50_: Half cytotoxicity concentration.

**Table 3 pharmaceutics-11-00448-t003:** Overview of the different polymers and hydrogels as a delivery system for AMPs.

Peptide/ Antibacterial Activity	Sequence	Delivery System	Bacterial Strain	Findings
Esculentin-1 Gram-negative	GIFSKLAGKKIKNLLISGLKG	PLGA NPs [48]	*P. aeruginosa*	Esculentin-1-loaded PLGA NPs displayed prolonged in vitro antimicrobial activity against *P. aeruginosa*, compared with the free peptide. Conjugated peptides led to an important reduction in the number of *Pseudomona* cells in the lung compared with the bacterial clearance employing the corresponding peptides in their soluble free form.
K4 broad-spectrum	KKKKPLFGLFFGLF	PLGA NPs [49]	*S. aureus* *P. aeruginosa*	K4 peptide and PLGA-K4 NPs killed ~75% and ~40% of *S. aureus* and ~50% and ~30% of *P. aeruginosa,* respectively.
Ultrashort AMP Gram-positive	RBRBR	Chitosan NPs [50]	*S. aureus* *MRSA strains*	RBRBR chitosan NPs were active against wild-type and the multidrug-resistant clinical isolated strains of Gram-positive bacteria. TRBRBR chitosan NPs presented antibiofilm activity.
Nisin Gram-positive	ITSISLCTPGCKTGALMGCNMKTATCHCSIHVSK	API-PUL nanofibers [51]	*S. typhimurium* *L.monocytogenes* *L. mesenteroides*	API-PUL nanofibers loaded 20 mg of nisin/mL. Microbial population reduction in apple juice, inactive *L. mesenteroides*, *S. Typhimurium,* and *L. monocytogenes* bacteria, while the nisine alone did not present antibacterial activity.
ε-PL broad-spectrum	21 to 35 l-lysine residues	PAA/PVA electrospun nanofibers [52]	*S. epidermidis* *S. aureus* *E. coli*	The differences in antibacterial efficiency between ε-PL-functionalized and non-functionalized fibers reached one order of magnitude after 14days for liquid cultures in contact with growing cultures.
Pac-525 broad-spectrum	KWRRWVRWI	AMP@PLGAMS@ Gln/chitosan/nHAp [53]	*S. aureus* *E. coli*	The inhibitory ratio of the 1-week-elution solution treated with polymer-loaded system was 94.61% and 95.08% against *E. coli* and *S. aureus,* respectively. As for the 4-week-elution solution, it was 68.26% and 77.36%, respectively.
β-amino acid-based peptidomimetic	(ACHC-β3hVal-β3hLys)3	Titanium surfaces with chitosan/hyaluronic acid polymer multilayers [54]	*S. aureus*	Improved prevention (up to 24 days) of biofilm formation on β-peptide-loaded coatings was achieved compared to uncoated substrates and films without the peptide. Release from the coatings took place over a 28-day period, and after 36 days, biofilm viability was reduced about 60% on coatings loaded with β-peptide compared to bare titanium. Minimal toxicity was observed against MC3T3-E1 cells.
Cecropin-melittin	KWKLFKKIGIGAVLKVLTTGLPALISC	Polymer surfaces [55]	*E. coli*	DBM-immobilized AMP presented similar antimicrobial activity after 5-day air exposure compared to the first day, while SAM-immobilized AMP had less antimicrobial activity the first day and no observable antimicrobial activity after 5 days. The surface-immobilized peptides killed bacteria by charge interaction and decrease in antibacterial activity is due to the loss of the peptides from the surface, maybe due to SAM decomposition, the DBM surface being more suitable.
*-*	ILPWRWPWWPWRR	Antiadhesive polymer brushes [56]	*S. aureus* *S. epidermidis* *P. aeruginosa*	Good antiadhesive and bactericidal properties were observed for coatings composed by PF127 polymer, PF127 modified with AMP, and PF127 modified with RGD in certain ratios, showing good tissue compatibility.
LL-37 derivative Broad spectrum	GFKRIVQRIKDFLRNLV	PEG [58]	*E. coli*	*E. coli* IC_50_ after 1 h was 40 ± 30 for the native peptide, which no longer showed antibacterial activity; 9 ± 5, 8 ± 4, and 8 ± 5 for the PEG conjugated analogue; and 20 ± 10, 20 ± 10, and 50 ± 20 for the methoxy PEG analogue, at 1, 6, and 24 h, respectively. Temporary masking of AMP arginine residues protected it against blood protease degradation and its bioactivity remained after a sustained release, which is beneficial for AMP-based therapies.
Aurein 2.2Δ3-cys	GLFDIVKKVVGALC	HPG [62]	*S. aureus* *S. epidermidis*	Aurein 2.2Δ3-cys antimicrobial activities, expressed as MIC, were 16 µg/mL for both *S. aureus* and *S. epidermidis*, while for or HPG-aurein 2.2Δ3-cys 2.5% conjugation ratio were 125 and 150 µg/mL, and for 5% conjugation ratio, 110 and 120 µg/mL, for *S.aureus* and *S.epidermidis*, respectively. After conjugation, antimicrobial activity decreased, so peptide density should be optimized to have activity without toxicity.
Cecropin A (CPA) Gram-negative	KWKLFKKIEKVGQNIRDGIIKAGPAVAVVGQATQIAK-NH2	PEG hydrogel surfaces (PEGSH/PEGDAE formulations) [66]	*S. sonnei* *E. coli*	CPA-functionalized hydrogels antimicrobial activity against *E. coli* was tested. The antimicrobial behavior of immobilized CPA depended on location variation in the peptide sequence and relationship between linker type and bactericidal activity.
CPA-K* Gram-negative	KWKLFKKIEK VGQNIRDGII KAGPAVAVVG QATQIAKK*–			
inverso-CysHHC10 Broad Spectrum	H-KRWWKWIRW-NH2	EGDA/PTMP [67]	*S. aureus* *S. epidermidis* *E. coli*	6-log reduction of bacteria of the 10 wt% AMP containing coating as compared to the blank hydrogel without AMP.
AP114 Gram-positive	GFGCNGPWNEDDLRCHNHCKSIKGYKGGYCAKGGFVCKCY	MAA26.5 and MAA60 microgels [68]	*E. coli* *P. aeruginosa*	Incorporated peptides can be protected from degradation by infection-related proteases at high microgel charge densities. MIC values revealed that no difference exists for *E. coli* when treated with AMP hydrogel or AMP alone. While, for *P. aeruginosa* strains, an improved MIC was observed for the DPK-060-loaded MAA26.5 microgels. For LL-37, a pronounced increase in MIC was observed as a consequence of encapsulation to the peptide into microgel.
LL-37 Broad spectrum	LLGDFFRKSKEKIGKEFKRIVQRIKDFLRNLVPRTES			
DPK-060 Broad spectrum	HKNKGKKNGKHNGWKWW			

MIC: Minimum inhibitory concentration; IC_50_: Half maximal inhibitory concentration.

**Table 4 pharmaceutics-11-00448-t004:** Overview of the different liposomes and liquid crystalline nanoparticles (LCNPs) as delivery systems for AMPs.

Peptide/ Antibacterial Activity	Sequence	Delivery System	Bacterial Strain	Findings
LL-37	LLGDFFRSKEKIGKEFKRIVQRIKDFLRNLVPRTES	Liposomes coated with PEG [85]	*Herpes simplex virus 1 (HIV-1)*	Lower cytotoxicity of LL-37 liposomes was found in comparison to indolicidin liposomes. Treatment with LL-37 alone resulted in a narrow therapeutic window, with antiviral activity EC_50_ = 18.7 μM and cytotoxicity CC_50_ = 37.3 μM. However, liposomal LL-37 with EC_50_ = 4.2 μM and CC_50_ = 43.8 μM presented a wider antiviral activity at lower concentrations.
Indolicidin	CILPWKWPWWPWRR			
Alyteserin-1c Gram-negative	GLKEIFKAGLGSLVKGIAAHVAS	Eudragit-coated liposomes [86]	*E. coli*	Increased antibacterial activity was observed after encapsulation and peptide chemical degradation could be prevented. MIC values for Alyteserin-1 (+2 and +5 peptide) were of 15.2 and 62.5 μM, while after coating with Eudragit liposomes, it was reduced to 1.25 and 5 μM, respectively.
Nisin Gram-positive	ITSISLCTPGCKTGALMGCNMKTATCHCSIHVSK	Pectin or polygalacturonic acid coated liposomes [87,89]	*L. monocytogenes* *L. innocua* *Listeria sp.*	The initial nisin release of coated liposomes was lower and more sustained during the first 30 h compared with that of non-coated, probably due to nisin interaction with the negatively-charged polysaccharides. Among the two coatings assayed, polygalacturonic liposomes maintained a higher antimicrobial activity after 14 days since the activities observed, first day, after 7 and 14 days, respectively were: 400, 400, and 200 AU/mL for non-coated liposomes; 800, 200, and 0 AU/mL for pectin-coated liposomes; 800, 400, and 200 AU/mL for polygalacturonic acid-coated liposomes.
		Chitosan or chondroitin sulphate coated liposomes [89]	*L. monocytogenes* *L. innocua* *Listeria sp.*	The incorporation of chitosan reduced bilayer thickness giving better-organized and more stable structures, which could be related with the better maintenance of antimicrobial activity observed. Initial antibacterial activity of liposomes was the same as for nisin alone (3200 AU/mL); however, nisin lost its activity after 6 h and bacteria grew back, while at 4 and 6 h, liposomes containing nisin reduced bacteria population to almost zero.
Apep10 Gram-positive	GLARCLAGTL	Chitosan coated liposomes [88]	*L. monocytogenes*	Bacterial-targeted delivery was achieved, since Apep10 was only released from the chitosan-coated liposomes in presence of the LLO secreted by *L. monocytogenes*, a toxin that leads to pore formation in liposome formulation. Release was regulated by the extent of bacterial contamination at initial stage.
Microcin J25 Gram-negative	GGAGHYPEYFVGIGTPISFYG	Dual-coated pectin and whey proteins (WPI) liposomes [90]	*S. enteritidis*	The coating process was optimized to improve the encapsulation efficiency and the protection of microcin against gastrointestinal digestion. Double-coated (pectin/WPI) liposomes showed a significant lower degradation of microcinJ25 than that obtained with single coated or non-coated liposomes after 2 h digestion. This formulation could be suitable for colon-targeted release.
AP114 Gram-positive LL-37 Broad spectrum DPK-060 Broad spectrum	GFGCNGPWNEDDLRCHNHCKSIKGYKGGYCAKGGFVCKCY LLGDFFRKSKEKIGKEFKRIVQRIKDFLRNLVPRTES HKNKGKKNGKHNGWKWW	Compositions: GMO or GMO/OA [93,94,95]	*S. aureus* *P. aeruginosa* *E. coli* *A. baumannii*	The antimicrobial effect of the peptide-loaded cubosomes was preserved (AP114) or sometimes even slightly enhanced (DPK-060) on *S. aureus* and *E. coli*. Cubosomes loaded with LL-37 displayed a loss in their broad-spectrum bacterial killing and were found to only have activity against Gram-negative strains.
p1025 Gram-positive	c-QLKTADLPAGRDETTSFVLV	Compositions: TTO, PPCA, PP and polycarbophil dispersion [96]	*S. mutans*	Protective effect by LLC over P1025 peptide. The conjugate preserved the anticaries and bioadhesive properties.

MIC: Minimum inhibitory concentration; IC_50_: Half maximal inhibitory concentration.

**Table 5 pharmaceutics-11-00448-t005:** Overview of dendrimers, cyclodextrins, and aptamers as delivery systems for AMPs.

Peptide	Sequence	Delivery System	Bacterial Strain	Findings
**AMP3** **Broad spectrum**	H-CRKWVWWRNR	MalG_2_(S(CH_2_)_2_N^+^Me_2_H·Cl^−^)_4_ [105]	*S. aureus* *E. coli*	Synergy studies showed an additive effect between carbosilane dendron and AMP3. Dendrons, AMP3, and their covalent conjugates can permeabilize bacterial membrane, causing significant morphological alterations and cellular integrity damages.
KR12	KRIVQRIKDFLR	2-hydroxypropyl-β-cyclodextrin [108]	*S. mutans* *A. actinomycetemcomitans* *P. gingivalis*	Antibacterial activity of the inclusion complex was enhanced, since MIC values were 7.8, 15.6, and 3.9 lg/mL for *S. mutans*, *P. gingivalis*, and *A. actinomycetemcomitans*, respectively. For KR12, all strains had an MIC of 7.8 lg/mL. Regarding toxicity, the complex had a lower hemolytic effect than KR12 alone, being, respectively, 48% and 73% for the maximum concentration assayed, as well as 10% and 30–40% at the lowest concentration. KR12 alone presented higher toxicity for fibroblasts (138% vs 104% LDH release), meaning that the cyclodextrin had a protective effect.
CM4	GRWKIFKKIEKVGQNIRDGIVKAGPAVAVVGQAATI	β-cyclodextrin [106]	*E. coli* *P. aeruginosa* *Aspergillus Niger* *Penicillium chrysogenum*	In vitro antimicrobial activity results for the complex were similar to those for CM4. In vivo studies against *P. aeruginosa* performed in mice showed that the mice treated with the complex had more viability (60%) than those treated with CM4 (20%) 12 h prior to *P. aeruginosa* infection, and after infection, both complex and CM4 alone protected from lung injury, with complex showing higher protection efficiency by abdominal treatment.
Alamethicin	XPXAXAQXVXGLXPVXXEQF	γ-cyclodextrin [107]	*L. monocytogenes*	While alamethicin was not able to inhibit bacterial growth in aqueous medium, the complex exhibited significant antimicrobial activity, which is dependent on γ-cyclodextrin/alamethicin molar ratio. The best antimicrobial activity was found for the γ-cyclodextrin/alamethicin 5:1 mole ratio with a MIC value of 2.1875 mg/mL (4.1563 mg/mL for 10:1 complex).
A3-APOHis	RPDKPRPYLPRPRPPRPVRHHHHHH	AuNPs conjugated with DNA aptamer [111]	*S. Typhimurium*	Conjugates enhanced the bactericidal activity of A3-APOHis against intracellular bacteria by efficiently delivering it through the plasma membranes of mammalian cells and producing disruption of bacterial membrane. 100% survival of infected mice treated with the complex was observed, being the viable *S. Typhimurium* cells in organs ∼93–98% reduced compared with those of mice treated with buffer, A3-APOHis, AuNP-AptHis, or another peptide conjugate assayed.
HPA3PHis	AKKVFKRLPKLFSKIWNWKHHHHHH	AuNPs conjugated with DNA aptamer [112]	*V. vulnificus*	Intravenous injection of the complex led to a complete inhibition of *V. vulnificus* colonization in *V. vulnificus*-infected mice by bacterial membrane disruption, leading to 100% survival rate among the treated mice, whereas all infected mice injected with buffer, AuNP-AptHis, or HPA3PHis died before 42 h after infection.

MIC: Minimum inhibitory concentration.

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
