# Peer review of "Nanosystems as Vehicles for the Delivery of Antimicrobial Peptides (AMPs)"

_pharmaceutics, 2019, doi:10.3390/pharmaceutics11090448_

Round 1
Reviewer 1 Report
In this article, authors reviewed recent delivery of antimicrobial peptides by nanomaterials. Authors summarized strategies and research results in recent studies during the last five years. The aim of this review that focused on the latest information makes it interesting article and will give beneficial information the readers.
However, the following points should be considered to help readers understand. If the following points are improved, the readers will be more benefited from this interesting article.
1) I recommend authors to add tables summarize delivery of AMP by nanomaterials. It would be preferable that the tables show information including 1) AMP names, 2) AMP sequences 3) Nanomaterials as delivery carriers for AMPs, 4) Bacterial strains used in studies, 5) Advantages of AMP with nanomaterials in comparison with AMP alone, and 6) Mechanism of action of nanomaterials showing advantages. The tables can be shown individually in the end of each section or integrated in one large table. If these tables (or table) are added, it will be great help to readers understand.
2) A font style used in all figures is difficult to read due to mixture of capital and small letters. Please change present font style to formal font.
3) In line 84, 146, 147 and 151, authors should not abbreviate names of bacterial strains due to first appearance.
4) In line 81, 96, 117 and 260, species names of bacterial strains should be shown.
5) In line 132, a space should be inserted in between ‘45’ and ‘polymer’.
6) Line 231 and 271, authors should write ‘et al’ in italic letter.
7) In line 242, authors should write ‘via’ in italic letter.
Author Response
Response to Reviewer: 1
1) I recommend authors to add tables summarize delivery of AMP by nanomaterials. It would be preferable that the tables show information including 1) AMP names, 2) AMP sequences 3) Nanomaterials as delivery carriers for AMPs, 4) Bacterial strains used in studies, 5) Advantages of AMP with nanomaterials in comparison with AMP alone, and 6) Mechanism of action of nanomaterials showing advantages. The tables can be shown individually in the end of each section or integrated in one large table. If these tables (or table) are added, it will be great help to readers understand.
As suggested referee 1 and also the referee 2, tables with the information required have been included in the text.
2) A font style used in all figures is difficult to read due to mixture of capital and small letters. Please change present font style to formal font.
The font style used in all figures have been changed
3) In line 84, 146, 147 and 151, authors should not abbreviate names of bacterial strains due to first appearance.
All the names of the bacteria have been checked and named following the referee suggestion.
4) In line 81, 96, 117 and 260, species names of bacterial strains should be shown.
5) In line 132, a space should be inserted in between ‘45’ and ‘polymer’.
6) Line 231 and 271, authors should write ‘et al’ in italic letter.
7) In line 242, authors should write ‘via’ in italic letter.
All the grammatical and spelling mistakes have been corrected

Reviewer 2 Report
The manuscript entitled “Nanomaterials as Vehicles for the Delivery of Antimicrobial Peptides (AMP)" by Angela Martin-Serrano, et al. described a review of recent developed carriers for AMPs. These trials seem interesting. However, I found a lot of serious major flaws to be modified. Consequently, the reviewer concluded that this manuscript under present state is very primitive and does not meet the standard of pharmaceutics (IF 4.773).
Serious major flaws:
1. The title is not reflected to the contents. Are Cyclodextrins, Aptamers and AMPs themselves, nanomaterials??? Moreover, the manuscript includes an application of AMPs (“AMPs for cancer therapy”). The title should be modified.
2. Arrangement of chapters should be reconsidered. For example, in the “recent trends” chapter, deliver systems and an application of AMPs are mixed. They should be separated in different chapters. These deliver systems such as “AMP hybrids” and “Antimicrobial peptide dendrimers” should be combined with “AMPs as vehicles” in a chapter before the “recent trends”. Moreover, the sections before the “recent trends” such as “polymers” include “recent trends” because they referred ref. 39, 40, which are published in 2019, for example.
I think “AMPs for cancer therapy” section should be in an independent chapter. And this chapter should include more applications using AMPs in recent years.
3. Related to 2, the vehicles should be categorized by sizes or concepts/systems. And each category should be in each chapter. For example, nanostructures, polymers, middle sized molecules, small molecules, systems using AMPs, ???
4. The descriptions should be more scientific and concrete. Please express the results numerically as much as possible.
For example, what are IC50 or MIC values? How much did those improve? (What times?)
5. Please compare all vehicles and systems, and show their merits and demerits. A table showing summary of merits and demerits in vehicles and systems is needed to be easily understandable for the readers.
Minor points:
1. To be easily understandable for the readers, AMPs in figures should be changed because they do not seem a peptide but a protein.
2. There are a lot of errors to be modified in the main text. For example, at line 108, Aich and et al. (2015), “and” should be deleted. “et al.” should be in Italic.
3. Please check authors, titles, volumes, pages, etc. in References carefully. There are a lot of errors to be modified. For example, in the journal names of ref 105, 113 and 116, all words begin with a capital letter.
Author Response
Response to Reviewer: 2
Serious major flaws:
1. The title is not reflected to the contents. Are Cyclodextrins, Aptamers and AMPs themselves, nanomaterials??? Moreover, the manuscript includes an application of AMPs (“AMPs for cancer therapy”). The title should be modified.
Thanks to reviewer for his/her suggestion, we agree with it and accordingly we have changed the title to:
Nanostructures as Vehicles for the Delivery of Antimicrobial Peptides (AMP).
Also, the application of AMP for cancer therapy have been exclude of this review. We agree with the referee that perhaps it is more appropriate to treat it in a different paper.
2. Arrangement of chapters should be reconsidered. For example, in the “recent trends” chapter, deliver systems and an application of AMPs are mixed. They should be separated in different chapters. These deliver systems such as “AMP hybrids” and “Antimicrobial peptide dendrimers” should be combined with “AMPs as vehicles” in a chapter before the “recent trends”. Moreover, the sections before the “recent trends” such as “polymers” include “recent trends” because they referred ref. 39, 40, which are published in 2019, for example.
I think “AMPs for cancer therapy” section should be in an independent chapter. And this chapter should include more applications using AMPs in recent years.
3. Related to 2, the vehicles should be categorized by sizes or concepts/systems. And each category should be in each chapter. For example, nanostructures, polymers, middle sized molecules, small molecules, systems using AMPs, ???
Thanks to the reviewer for his/her suggestion in points 2 and 3. Accordingly to these recommendations the structure of the manuscript has been reorganized (all changes are indicated in yellow).
The new classification is as follows:
1. Nanomaterials: metal nanoparticles, carbon nanotubes, liposomes, crystalline liquids, dendritic systems.
2. Polymer systems: polymers, hydrogels
3. Cyclodextrins
4-Aptamers conjugates
5- AMP as vehicles: in this section we have combined the old distribution “AMP hybrids”, “Antimicrobial peptide dendrimers” and “AMPs as vehicles”
4. The descriptions should be more scientific and concrete. Please express the results numerically as much as possible. For example, what are IC50 or MIC values? How much did those improve? (What times?)
Tables have been included at the end of each section where the most relevant data from each publication are collected in order to facilitate reading and obtain the most relevant results quickly. reviewing again all the publications, not all give MIC or IC50, so we added in the table the information that in our opinion is more relevant. In those publications in which if such data are given, these have been included
5. Please compare all vehicles and systems, and show their merits and demerits. A table showing summary of merits and demerits in vehicles and systems is needed to be easily understandable for the readers.
A table (table 1) has been include in the manuscript that show some of advantages and disadvantages of each system in drug delivery. Also in each case have include a bibliographic reference
Minor points:
1. To be easily understandable for the readers, AMPs in figures should be changed because they do not seem a peptide but a protein.
2. There are a lot of errors to be modified in the main text. For example, at line 108, Aich and et al. (2015), “and” should be deleted. “et al.” should be in Italic.
3. Please check authors, titles, volumes, pages, etc. in References carefully. There are a lot of errors to be modified. For example, in the journal names of ref 105, 113 and 116, all words begin with a capital letter.
All minor points have been corrected.

Reviewer 3 Report
This review provides a useful and deep overview on the findings concerning the use of nanotechnology to efficiently deliver antimicrobial peptides. Main experimental strategies and methodologies are clearly described, by fully highliting adavatages and limitations of each of them. On the basis of these observations, the reviewer expresses an overall positive judgment, completely favorable to publication, upon minor spell checking along the whole manuscript.
Author Response
minor spell checking have been corrected
Round 2
Reviewer 2 Report
The manuscript entitled “Nanostructures as Vehicles for the Delivery of Antimicrobial Peptides (AMP)" by Angela Martin-Serrano, et al. described a review of recent developed carriers for AMPs. These trials seem interesting and the revised manuscript are improved some. However, the authors DID NOT modify all my points. I still found some serious major flaws as below, which would need a revision of the manuscript to fully address:
Serious major flaws:
1. Categorization of the vehicles should be reconsidered. It is hard for the readers to understand these categories because vehicles were not categorized by same criteria / concept. For example, “Dendritic system” is not nanomaterials. On the other hand, “hydrogel” seems nanomaterials but not was categorized in. To begin with, polymers, nanomaterials and cyclodextrin are in different classification concept / criteria.
As I pointed out at the 1st report, the authors should categorize them by sizes or concepts/systems (by SAME criteria / concept).
2. Conclusions should be modified. This seems 1st version...
3. To be easily understandable for the readers, graphical abstract should be modified because it is hard to understand a grey quadrangle having a green arrow is a road?
Minor points:
There are STILL lots of errors to be modified in the text. I show some, not all.
1. At line 83, “The so formed peptide/AuNP conjugate”, what is “so”?
2. Ref. 98 seems missing in the main text.
3. Please RECHECK authors, titles, volumes, pages, etc. in References carefully. There are STILL a lot of errors to be modified.
For example, The journal names of Ref.14, 18, 26, 32, 36、49, 78, 97, 118 need some periods.
The journal name of Ref. 98 is in capital letters.
In the title of ref. 39, all words should begin with a capital letter.
The title of Ref. 45 is in capital letters.
A line break should be deleted in Ref. 69.
Ref. 47 is not in English? It should be noticed?
......
Author Response
Serious major flaws:
Categorization of the vehicles should be reconsidered. It is hard for the readers to understand these categories because vehicles were not categorized by same criteria / concept. For example, “Dendritic system” is not nanomaterials. On the other hand, “hydrogel” seems nanomaterials but not was categorized in. To begin with, polymers, nanomaterials and cyclodextrin are in different classification concept / criteria.As I pointed out at the 1st report, the authors should categorize them by sizes or concepts/systems (by SAME criteria / concept).
Thanks again to the referee for this suggestion. We agree with him/her on the convenience of classifying the different types of nanosystems to facilitate the understanding of potential readers. We also agree that it was confusing to classify these nanosystems into materials or non-materials in the way that was written in our last version. For these reasons and following suggestion´s referee we have stablished a new classification of the nanosystems described in this paper.
Our choice has been to classify these systems according to the nature of the skeleton or scaffold that they contain, thus dividing them into inorganic and organic systems. Finally, we have included a third group to discuss the use of AMP as vehicles by themselves. A sentence has been included in the text to refer the new classification. (pag 3, line 71)
These classification (organic/inorganic material) have been used for other authors in previous articles or reviews and we think it might also be a good classification for the systems included in this review. Some references supporting this classification are: ((i) Tatiparti, K., S. Sau, et al. (2017). "siRNA Delivery Strategies: A Comprehensive Review of Recent Developments." Nanomaterials 7(4): 77. (ii) Shah, S. (2016). "The nanomaterial toolkit for neuroengineering." Nano Convergence 3(1): 25. (iii) Baranwal, A., A. Srivastava, et al. (2018). "Prospects of Nanostructure Materials and Their Composites as Antimicrobial Agents." Frontiers in microbiology 9(422))
Conclusions should be modified. This seems 1st version...
We have modified the conclusions trying to give also a perspective of the future that can present these nanosystems for the delivey of AMPs and the importance of nanotechnology to increase the performance of these AMPs as antibacterial agents.
To be easily understandable for the readers, graphical abstract should be modified because it is hard to understand a grey quadrangle having a green arrow is a road?
A grey quadrangle in the graphical abstract represents a Scalextric track, because a car built with mechanical parts has been used to represent the vehiculization of the antimicrobial peptides.
The green arrow represents the path that this vehicle must follow to transport the AMP and access the bacteria membrane to exert its antibacterial effect.
We hope that we this explanation the referee accepts this graphical abstract. Nevertheless if he/she considers that still can be difficult to interpretate we will try to change it. Nevertheless, we think that this picture gives a fast idea of how construction of a specific vehicle, using in this case nanotechnology, could be a good approach for the delivery AMPs to their place of action
The word “Nanostructures” has been changed by “Nanosystems” that is more appropriated to describe all the systems collected in this review
Minor points:
There are STILL lots of errors to be modified in the text. I show some, not all.
At line 83, “The so formed peptide/AuNP conjugate”, what is “so”?“so formed” have been eliminated to the text (pag 4, line 87) and we have tried to correct the rest of errors in the text.
Ref. 98 seems missing in the main text.The reference is not missing, is enclosed in the polymer conjugates references [95-101] (section 2.4 Polymers, pag 10, line 248).
Please RECHECK authors, titles, volumes, pages, etc. in References carefully. There are STILL a lot of errors to be modified.The bibliography has been checked.

Round 3
Reviewer 2 Report
The manuscript entitled “Nanosystems as Vehicles for the Delivery of Antimicrobial Peptides (AMP)" by Angela Martin-Serrano, et al. described a review of recent developed carriers for AMPs. These trials seem interesting and the revised manuscript are improved some. However, the authors DID NOT modify all my points. I still found some major flaws as below, which would need a revision of the manuscript to fully address, to meet the standard of “Pharmaceutics” (IF 4.773) :
Major points:
1. In figure 1, what illustration is it between "BACTERIA" illustration and "MEMBRANE DISRUPTION" illustration (upon the blue arrow)? Besides it seems the old version AMP is included.
2. Although I now agree to your classification (organic/inorganic nanosystems, AMP themselves), I do not agree to your classification in organic nanosystems. For example, cyclodextrin is not a polymer but an oligomer. Liposomes were so-called “lipid polymers”. Generally, we do not call dendrimers polymers, etc. (In your conclusion, you also DID describe “cyclodextrins or aptamers”, NOT “polymers”...)
Minor points
Please RECHECK authors, titles, volumes, pages, etc. in References carefully. There are STILL a lot of errors to be modified. I show some, not all.
Ex. 1 All words begin with a capital letter in title? If so, Ref 13, 16, 17, etc. should be modified. (Please unify a style.)
Ex. 2 In Ref. 124, the page is missing?
Author Response
Major points:
In figure 1, what illustration is it between "BACTERIA" illustration and "MEMBRANE DISRUPTION" illustration (upon the blue arrow)? Besides it seems the old version AMP is included.By reviewing Figure 1, we agree with the reviewer and would like to thank him/her for the observation. For this reason, the figure has been modified to avoid confusion for the reader. In addition, in the text where the possible modes of action of MPAs are mentioned (page 2, lines 42-44), a note to Figure 1 has been included (page 2, line 47). Also, the AMP has been changed and now is the same in all the figures. Nevertheless if he/she considers that still can be difficult to interpretate we will try to change it again.
Although I now agree to your classification (organic/inorganic nanosystems, AMP themselves), I do not agree to your classification in organic nanosystems. For example, cyclodextrin is not a polymer but an oligomer. Liposomes were so-called “lipid polymers”. Generally, we do not call dendrimers polymers, etc. (In your conclusion, you also DID describe “cyclodextrins or aptamers”, NOT “polymers”...)
Thanks again to the referee for this suggestion. We agree with him/her on the convenience to restructure the classification of organic nanosystem because dendrimers, cyclodextrins and aptamers not should be considered properly as polymers.
Regarding to lipid-based systems we prefer to consider them under a different epigraph to avoid possible confusions. In this revised version of the paper we have classified them as Lipid based systems. The term “lipid polymers” referred by the reviewer is used to hybrid systems where lipid systems are conjugated with polymers as shown in the following bibliographic citations. [1-4] including in this letter.
We have also included in this category of lipid-based systems to Liquid crystalline particles since the ones referred in this review are constituted by lipids (cubosomes and hexoxomas).
Date, T.; Nimbalkar, V.; Kamat, J.; Mittal, A.; Mahato, R. I.; Chitkara, D. Lipid-polymer hybrid nanocarriers for delivering cancer therapeutics. 2018, 271,60-73. Seedat, N.; Kalhapure, R. S.; Mocktar, C.; Vepuri, S.; Jadhav, M.; Soliman, M.; Govender, T. Co-encapsulation of multi-lipids and polymers enhances the performance of vancomycin in lipid–polymer hybrid nanoparticles: In vitro and in silico studies. 2016, 61,616-630. García-Pinel, B.; Porras-Alcalá, C.; Ortega-Rodríguez, A.; Sarabia, F.; Prados, J.; Melguizo, C.; López-Romero, J. M. Lipid-based nanoparticles: application and recent advances in cancer treatment. Nanomaterials 2019, 9, (4),638. Dave, V.; Tak, K.; Sohgaura, A.; Gupta, A.; Sadhu, V.; Reddy, K. R. Lipid-polymer hybrid nanoparticles: Synthesis strategies and biomedical applications. 2019, 160,130-142.
Finally, we propose the subsequent division and we hope that this new subdivision will satisfy the reviewer opinion. Nevertheless, if still there is some doubt we could try to clarify it.
Organic Nanosystems2.1. Polymer Systems
2.1.1. Polymers
2.1.2. Hydrogels
2.2 Lipid based systems
2.2.1 Liposomes
2.2.2 Liquid crystalline particles
2.3 Dendritic systems
2.4 Cyclodextrins
2.5 Aptamer conjugates
Minor points:
Please RECHECK authors, titles, volumes, pages, etc. in References carefully. There are STILL a lot of errors to be modified. I show some, not all.
Ex. 1 All words begin with a capital letter in title? If so, Ref 13, 16, 17, etc. should be modified. (Please unify a style.)
Ex. 2 In Ref. 124, the page is missing? There are STILL lots of errors to be modified in the text. I show some, not all.
The bibliography has been checked and we have homogenised the style in the titles of the publications.

Round 4
Reviewer 2 Report
The manuscript entitled “Nanosystems as Vehicles for the Delivery of Antimicrobial Peptides (AMP)" by Angela Martin-Serrano, et al. described a review of recent developed carriers for AMPs. These trials seem interesting and the revised manuscript are improved. However, I still found some minor points as below, which would need a revision of the manuscript to fully address:
Minor points:
At line 23, if you numbered “Conclusions”, “Introduction” should be numbered. (Chapter numbering should begin from “Introduction”.)Besides, the period should be deleted? (“Introduction.” --> “Introduction”)
In Table 1 the order of drug carriers should be modified as the main text.
A single space should be needed between numerical values and the units. For example, at the row of “indolicidin…” in Table 2, “0.02ug/mL” ---> “0.02 ug/mL”. There are a lot of errors in the text.
At line 137, “Polymer” --> “Polymers”
At the row of “LL-37 derivative…” in Table 3 units are missing. For example, 40+-30 ???
Author Response
Minor points:
At line 23, if you numbered “Conclusions”, “Introduction” should be numbered. (Chapter numbering should begin from “Introduction”.) Besides, the period should be deleted? (“Introduction.” --> “Introduction”)We have numbered all the epigraphs to give uniformity to the text and deleted the period after the final word.
In Table 1 the order of drug carriers should be modified as the main text.
The table has been modified accordingly to the reviewer suggestion.
A single space should be needed between numerical values and the units. For example, at the row of “indolicidin…” in Table 2, “0.02ug/mL” ---> “0.02 ug/mL”. There are a lot of errors in the text.
We have corrected the text accordingly to the reviewer suggestion.
At line 137, “Polymer” --> “Polymers”
The word has been corrected.
At the row of “LL-37 derivative…” in Table 3 units are missing. For example, 40+-30 ???
The units (mM) have been included.
